# Enantioselective functionalization at the C4 position of pyridinium salts through NHC catalysis

Hangyeol Choi [1,2], Gangadhar Rao Mathi[1,2], Seonghyeok Hong[1,2] & Sungwoo Hong [1,2✉]

A catalytic method for the enantioselective and C4-selective functionalization of pyridine derivatives is yet to be developed. Herein, we report an efficient method for the asymmetric β-pyridylations of enals that involve N-heterocyclic carbene (NHC) catalysis with excellent control over enantioselectivity and pyridyl C4-selectivity. The key strategy for precise stereocontrol involves enhancing interactions between the chiral NHC-bound homoenolate and pyridinium salt in the presence of hexafluorobenzene, which effectively differentiates the two faces of the homoenolate radical. Room temperature is sufficient for this transformation, and reaction efficiency is further accelerated by photo-mediation. This methodology exhibits broad functional group tolerance and enables facile access to a diverse range of enantioenriched β-pyridyl carbonyl compounds under mild and metal-free conditions.

[1] Department of Chemistry, Korea Advanced Institute of Science and Technology (KAIST), Daejeon 34141, Korea. [2] Center for Catalytic Hydrocarbon Functionalizations, Institute for Basic Science (IBS), Daejeon 34141, Korea. ✉email: hongorg@kaist.ac.kr

The pyridine moiety is an important structural unit found in many bioactive molecules, functional materials, and naturally occurring compounds[1,2]. Consequently, great effort has been devoted to functionalizing pyridine-containing compounds[3–9]. Recently, the photoinduced heteroarene modification has proven to be a powerful strategy for dealing with elusive radical transformations under mild conditions[10–16]. Despite the remarkable achievements of the Minisci-type radical-addition reaction, precise control over the regiochemical outcome of such a reaction has proven to be challenging owing to the presence of multiple reactive sites, which limits the site-selective applicability of this approach. Moreover, there is an increasing drive for higher degrees of three-dimensionality in pyridine-derived molecules for exploring chemical space, which necessitates the asymmetric installation of chiral centers adjacent to the pyridine core. Hou and collaborators reported a scandium-catalyzed enantioselective C–H alkylation of pyridines at the C2 position (Fig. 1a)[17]. Recently, the radical-mediated asymmetric approach has provided exciting strategies for the enantioselective syntheses of pyridines bearing adjacent stereogenic centers. The Phipps group reported the enantioselective additions of α-aminoalkyl radicals to the C2-positions of pyridines by dual photoredox and chiral phosphoric acid catalysis (Fig. 1a)[18,19]. Subsequently, Studer et al. reported an asymmetric three-component Minisci-type reaction using α-aminoalkyl radicals generated in situ[20]. High levels of asymmetric induction in this platform were achieved by exploiting favorable two-point interactions between protonated pyridines, phosphoric acid, and α-aminoalkyl radicals. Therefore, the scope of this approach is primarily confined to the *ortho*(C2)-selective radical addition to the pyridine core, and a catalytic method for the enantioselective and C4-selective functionalization of pyridine derivatives is yet to be developed. In this context, there remains a high demand for a new strategy that enables pyridine derivatives to be asymmetrically functionalized at the C4 position.

N-Heterocyclic carbene (NHC) catalysis, a versatile organocatalytic process, has attracted significant interest from the synthetic community[21–27]. In particular, the development of NHC-catalyzed umpolung chemistry involving Breslow intermediates has provided unique approaches for the enantioselective construction of C–C bonds[28–30]. In recent years, single-electron NHC catalysis has emerged as a versatile tool for accessing new modes of radical reaction that were previously difficult to achieve[31–39]. Inspired by a

biological system, Fukuzumi[40] and Studer[41] disclosed NHC-mediated radical reactions of Breslow intermediates involving two single-electron-transfer (SET) processes. Since these pioneering studies, important contributions that enable the direct assembly of functional molecules with broad utility have been made to this field. Recently, Ohmiya and coworkers reported that the NHC-catalyzed SET process has important potential for carbon–carbon bond formation through radical–radical coupling[42–46]. Moreover, asymmetric radical reactions under chiral NHC catalysis offer unique opportunities for unconventional approaches to the synthesis of chiral molecules. The groups of Rovis[47,48] and Chi[49] demonstrated the chiral NHC-catalyzed single-electron oxidation of NHC-bound homoenolates using nitroarenes for the enantioselective syntheses of β-hydroxy esters. Despite significant achievements in NHC-catalyzed radical chemistry, the enantioselective coupling reaction between a homoenolate equivalent and a heteroarene remains a long-standing challenge but represents a straightforward approach that rapidly accesses value-added chiral β-heteroaryl carbonyl compounds. The requisite electron acceptors must be sufficiently electron-poor to chemically enable homoenolate oxidation[50,51].

Herein, we describe an NHC-catalyzed radical reaction that converts various enals to value-added chiral β-pyridyl esters with excellent control over absolute stereochemistry and pyridyl C4-selectivity. As outlined in Fig. 1c, a pyridinium salt behaves as an effective oxidant for the SET oxidation of an enal under NHC catalysis, thereby enabling a radical-mediated transformation. The homoenolate radical formed in situ by the SET oxidation of a pyridinium salt readily adds to the C4 position of the pyridine core and impart chiral control.

## Results

**Reaction discovery and optimization**. We propose that Breslow intermediate anion ($E^{o} = -1.7 - -1.8$ V vs. SCE)[39] has a sufficient reduction potential for a pyridinium salt ($E^{o} = -0.75$ to $-0.82$ V vs. SCE)[16]. At this juncture, we reasoned that Breslow intermediates would serve as efficient SET reductants for electron-deficient pyridinium salts to generate the corresponding homoenolate radicals. Intermolecular coupling of the homoenolate radical and the pyridinium salt would then forge a new C–C bond, with the resulting amidyl radical serving as an SET electron acceptor from the homoenolate to promote a chain process.

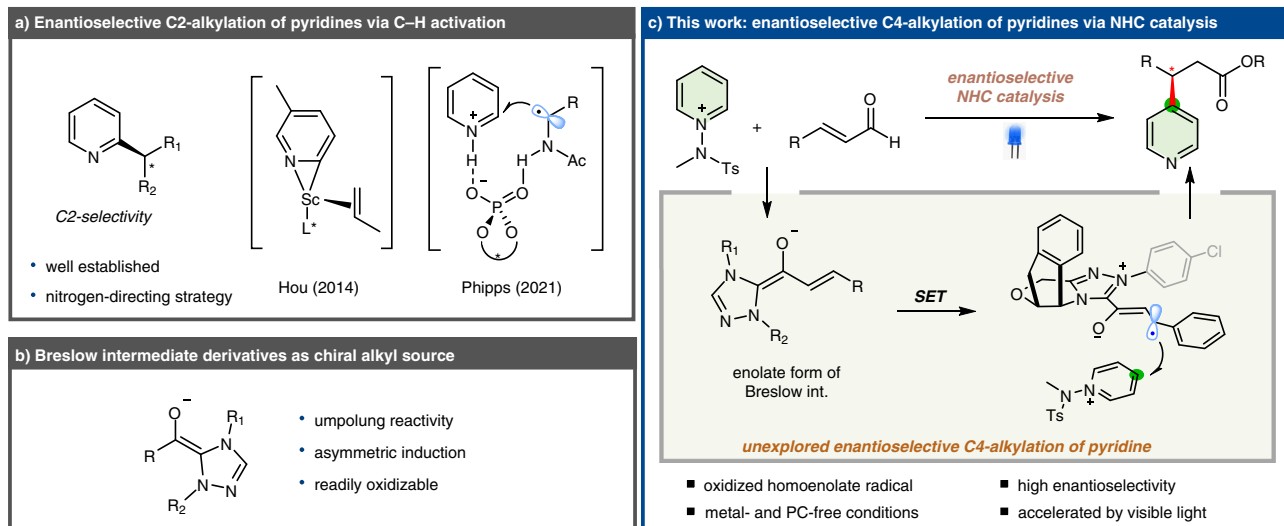

**Fig. 1 Design plan. a** Enantioselective C2-alkylation of pyridines. **b** Breslow intermediate derivatives as chiral alkyl source. **c** Enantioselective NHC-catalyzed β-pyridylation of enals with pyridyl C4-selectivity.

**Table 1 Optimization of the reaction conditions.**

| Entry | NHC | Solvent (M) | Light | Yield (%)[a] | er[b] |
|---|---|---|---|---|---|
| 1 | 5a | PhMe (0.05) | dark | 51 | – |
| 2 | 5b | PhMe (0.05) | dark | 24 | 77:23 |
| 3 | 5c | PhMe (0.05) | dark | 26 | 84:16 |
| 4 | 5b | PhMe (0.05) | blue LED | 47 | 78:22 |
| 5 | 5c | PhMe (0.05) | blue LED | 46 | 84:16 |
| 6 | 5d | PhMe (0.05) | blue LED | 55 | 85:15 |
| 7 | 5e | PhMe (0.05) | blue LED | 57 | 80:20 |
| 8 | 5f | PhMe (0.05) | blue LED | 58 | 89:11 |
| 9 | 5f | CHCl₃ (0.05) | blue LED | 10 | 77:23 |
| 10 | 5f | EtOAc (0.05) | blue LED | 16 | 78:22 |
| 11 | 5f | HFB (0.05) | blue LED | 69 | 94:6 |
| 12 | 5f | HFB (0.025) | blue LED | 67 | 96:4 |
| 13[c] | 5f | HFB (0.025) | blue LED | n.d. | – |
| 14[d] | 5f | HFB (0.025) | blue LED | n.d. | – |
| 15 | 5f | HFB (0.025) | dark | 53 | 96:4 |

Reaction conditions: **1a** (0.05 mmol), **2a** (2.0 equiv), NaOPiv•H₂O (1.0 equiv), NHC catalyst (15 mol%), MeOH (12.5 equiv) in solvent irradiated by blue LEDs (440 nm, 10 W) at rt under Ar for 14 h.
[a]Yields were determined by ¹H NMR analysis.
[b]Enantiomeric ratios were determined by HPLC.
[c]Without NHC catalyst.
[d]Addition of TEMPO (3 equiv). n.d. not detected.

With these mechanistic considerations in mind, we first examined our proposed strategy by monitoring the reactivities of pyridinium salt **1a** and enal **2a**, as shown in Table 1. The β-pyridyl ester product **3a** was obtained in 24% yield with an encouraging enantiomeric ratio (er) using a degassed solution of triazolium NHC precursor **5b** in toluene. Switching the NHC catalyst from **5b** to **5c** resulted in an improved er (84:16) and a similar yield (entry 3). Further studies revealed that irradiation with visible light facilitated more efficient conversion (entries 4 and 5). We observed that mixing the pyridinium salt[14,52–55] with pivalate resulted in a significant shift in the intensity of the UV/Vis absorption peak, suggestive of the formation of a photoactive electron donor–acceptor (EDA) complex (see Supplementary Information for details)[56–63]. Therefore, the intermediacy of the EDA complex further facilitates SET to generate amidyl radicals when irradiated by visible light, thereby promoting the radical chain pathway[49,64–69]. Screening chiral catalysts revealed that NHC scaffold **5f** provided the product with improved reactivity and enantioselectivity (entry 8); therefore, **5f** was selected for further optimization. Among the solvents screened, we found that hexafluorobenzene (HFB) provided higher enantioselectivity probably by establishing electrostatic interactions with the homoenolate during the radical addition to pyridine (entries 11 and 12)[70]. We were pleased to discover that excellent enantioselectivity (96:4, er) was obtained, together with very high regioselectivity (>20:1) for the pyridyl C4 position under our conditions (entry 12). Control experiments established the critical role of the NHC catalyst, as no desired reaction was observed in the absence of an NHC catalyst (entry 13). The reaction was completely inhibited by the addition of the 2,2,6,6-tetramethylpiperidine-1-oxyl radical (TEMPO), which indicates that this reaction proceeds via a radical-mediated mechanism (entry 14).

**Substrate scope studies.** With the optimal reaction conditions in hand, the substrate scope was next investigated to demonstrate the generality of this transformation. As shown in Fig. 2, a variety of α,β-unsaturated aldehydes serve as competent substrates under standard reaction conditions. For example, we were pleased to find that various electron-donating substituents at the *para*-position of the aryl ring, such as methyl (**3b**), methoxy (**3c**), and phenyl (**3d**), are well tolerated, with good-to-excellent ers obtained. Aldehydes bearing electron-deficient groups on the aryl ring are also suitable substrates, although slightly lower ers were observed (**3e**–**3j**). In addition, *meta*-substituted aryl enals were all found to be suitable substrates, irrespective of the electronic and steric effect of the substituent (**3k**–**3n**). A broad range of *ortho*-substituted enals, such as those bearing methyl (**3o**), methoxy (**3p**), chloro (**3q**), and bromo (**3r**) groups, readily participated in this reaction, with good enantioselectivities observed. Notably, the reactions of halogen-containing substrates readily produced the corresponding products, with the halogen moieties remaining intact. We also successfully used aldehydes bearing di-substituted aryl (**3s**), furan (**3t**), thiophene (**3u**), naphthalene (**3v**–**3x**) moieties as competent coupling partners with comparable reactivities

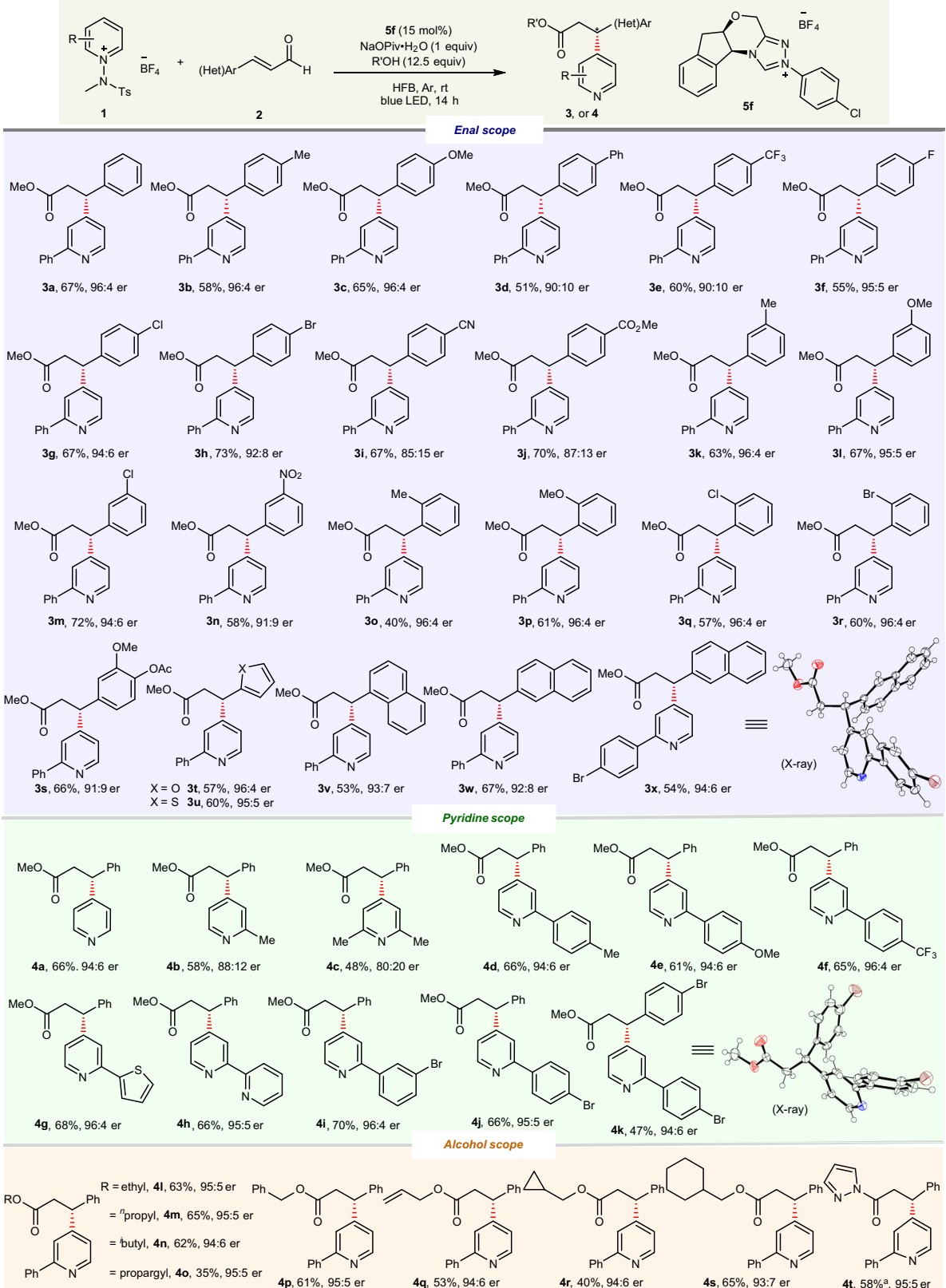

**Fig. 2 Substrate scope.** Reaction conditions: **1** (0.05 mmol), **2** (2.0 equiv), NaOPiv•H$_2$O (1.0 equiv), and **5f** (15 mol%) in HFB (2.0 mL) under light irradiation using blue LEDs (440 nm, 10 W) at rt under Ar for 14 h. [a]Pyrazole (2 equiv) was used.

**Fig. 3 Late-stage modification.** Standard reaction conditions.

and enantioselectivities. When β-alkyl enal was subjected to the standard reaction conditions, the desired product was not formed. We continued this study by evaluating the pyridinium salt scope. As shown in Fig. 2, this protocol is compatible with a broad range of N-amidopyridinium salts bearing either electron-donating or electron-withdrawing substituents.

The unsubstituted pyridinium salt was a viable substrate, and product **4a** was obtained with 94:6 er. The current method was amenable to the C2-substituted pyridinium salts bearing methyl (**4b** and **4c**) group. Likewise, various substituents and substitution patterns on the benzene rings of the pyridinium substrates were compatible with this reaction to furnish their respective products; methyl (**4d**), methoxy (**4e**), trifluoromethyl (**4f**), thiophene (**4g**), pyridine (**4h**), and bromo (**4i** and **4j**) groups were all tolerated. Importantly, enals were β-pyridylated exclusively at the C4 position of their pyridine cores in all cases, regardless of the substitution pattern. The absolute configurations of **3x** and **4k** were determined as (R) by X-ray diffraction analysis. The C3-substituted pyridinium salt was not compatible with this method, and the desired product was not obtained. We next showed that this transformation is applicable to a range of alcohol components. The reactions of linear (**4l** and **4m**), branched (**4n**), propargyl (**4o**), benzylic (**4p**), allylic (**4q**), alkyl alcohols with different chain lengths (**4r** and **4s**), and pyrazole (**4t**) were found to be effective with good-to-excellent enantioselectivities obtained.

To further demonstrate the broad applicability of the enantioselective transformation, we explored the late-stage modification of biorelevant molecules (Fig. 3). Pleasingly, pharmaceutically interesting compounds (estrone, L-phenylalanine, oxaprozin, and febuxostat derivatives) could be successfully employed in this enantioselective β-pyridylation, which afforded desired products (**6a**–**6d**).

**Control experiments and proposed mechanism.** Several control experiments were performed to gain more insight into the reaction mechanism. First, a mixture of **1a** and pyridine was reacted under our standard conditions to determine that the pyridinium salt is the only reactive radical acceptor in the reaction. The reaction proceeded only with pyridinium salt **1a**, whereas pyridine was not engaged in the reaction (Fig. 4a). Next, we observed that the presence of oxygen or galvinoxyl completely inhibited the reaction, indicating the radical-involved reaction pathway (Fig. 4b). A decreased product yield was observed with the addition of $H_2O$.

Based on the above observations, a plausible NHC catalytic mechanism is illustrated in Fig. 5. The enolate form of Breslow intermediate **I** undergoes direct SET with the pyridinium salt to give the amidyl radical and homoenolate-centered radical **II**. At this point, the resultant β-radical **II** selectively adds to the C4 position of the N-amidopyridinium salt, which is the enantioselectivity-determining step. The resulting radical **III** undergoes facile deprotonation and subsequent homolytic cleavage of the N–N bond to give the amidyl radical and NHC-bound intermediate **IV**. The alcohol then reacts with intermediate **IV** to regenerate the NHC catalyst and eventually yield the desired β-pyridylated ester. The in-situ-generated amidyl radical is expected to engage in an SET event with homoenolate **I**, which propagates the radical chain. Since light-promoted reactivity and rate acceleration were observed, we propose an alternative

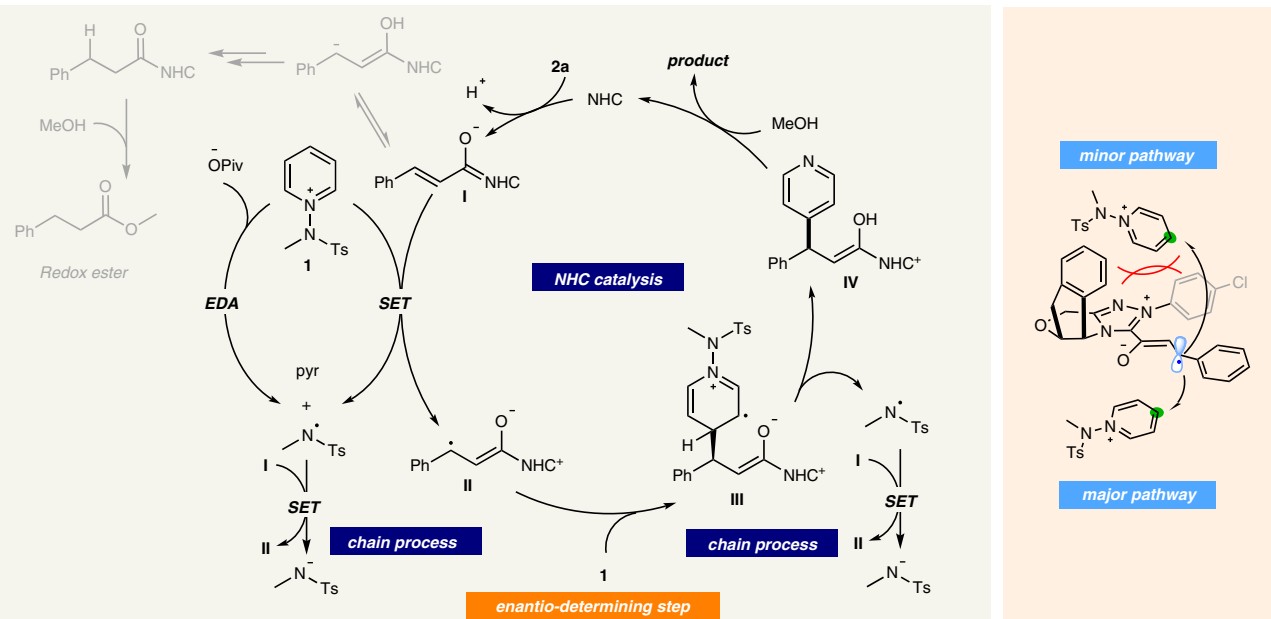

**Fig. 4 Control experiments. a** Reaction with mixture of **1a** and pyridine. **b** Reactions with the presence of O₂, galvinoxyl free radical, and H₂O.

**Fig. 5 Proposed reaction mechanism.** The asymmetric β-pyridylation was achieved through single-electron NHC catalysis with excellent enantioselectivity.

mechanistic platform that involves a photon-absorbing pyridinium−pivalate EDA complex, which triggers the formation of an amidyl radical when irradiated with visible light, in turn, initiating a new radical chain pathway. During the process, there are two conceivable unproductive reaction pathways. Firstly, the redox esterification process is a prototypical reaction of enals in NHC catalysis, which provides methyl cinnamate as a side product. Remarkably, when NHC catalyst **5f** was employed, the formation of the ester product was completely suppressed (see Supplementary Table 2 in the Supplementary Information). Secondly, the acyl radicals generated from the hydrogen atom abstraction (HAT) process between enals and amidyl radicals could add to pyridinium salts. Interestingly, the acylated pyridine was not detected in this reaction, and it is postulated that the amidyl radical reacts rapidly with **I** in a radical chain pathway.

## Discussion

In summary, we report a catalytic method for the asymmetric β-pyridylations of enals through single-electron NHC catalysis. The method delivers excellent control over enantio- and site-selectivity. Mechanistically, the reaction is believed to proceed through SET involving a chiral Breslow-intermediate-derived homoenolate and a pyridinium salt, resulting in a homoenolate radical that adds to the C4 position of the pyridine core. Notably, the use of hexafluorobenzene as a solvent is the key to achieving excellent enantioselectivities. This methodology exhibits broad application scope and provides a new retrosynthetic disconnection for the straightforward syntheses of a diverse range of enantioenriched β-pyridyl carbonyl building blocks under mild and metal-free conditions. Considering the ubiquity of carbonyl and pyridyl groups in chemical and biological molecular systems, the developed protocol represents a powerful method that can broadly be applied in synthetic and medicinal chemistry.

## Methods

**Representative procedure for the enantioselective β-pyridylation**. To a 16 mL test tube equipped with a Teflon-coated magnetic bar were added pyridinium salt (0.05 mmol), sodium pivalate hydrate (0.05 mmol), and NHC catalyst (0.0075 mmol). The test tube, anhydrous methanol, hexafluorobenzene degassed by freeze-pump-thaw, and cinnamaldehyde were placed into an argon-filled glovebox. In glovebox, stock solution of cinnamaldehyde (0.1 mmol) and MeOH (12.5 equiv) was prepared with 2.0 mL of hexafluorobenzene. The stock solution was added to the test tube. After adding solvent, the test tube was closed by cap and removed from the glovebox. The reaction mixture was stirred under 440 nm Kessil blue

LEDs (10 W, 25% intensity) at room temperature for 14 h. After reaction completion, the mixture was diluted and extracted with dichloromethane three times. The organic layer was dried over sodium sulfate and filtered. The resulting mixture was concentrated under reduced pressure and purified by flash column chromatography on silica gel (ethyl acetate: *n*-hexane = 1:4 or CH₂Cl₂:MeOH = 30:1) to obtain the desired product **3a** (67%, 10.6 mg).

## Data availability

Experimental procedure and characterization data of new compounds are available within Supplementary Information. Full crystallographic data of all structurally characterized compounds described herein are given in the Supplementary Information. Depository Number CCDC 2116151 (**3x**), 2116149 (**4k**), 2116154 (**3d**), 2116152 (**3e**), and 2116153 (**4d**) contain the supplementary crystallographic data for this paper. These data are provided free of charge by the joint Cambridge Crystallographic Data Centre and Fachinformationszentrum Karlsruhe Access Structures service www.ccdc.cam.ac.uk/structures.

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

## Acknowledgements

This research was supported financially by Institute for Basic Science (IBS-R010-A2). We thank Dr. Dongwook Kim (IBS) for XRD analysis.

## Author contributions

H.C., G.R.M., and Seonghyeok Hong performed the experiments. Sungwoo Hong directed the project. All authors contributed to the preparation of the manuscript.

## Competing interests

The authors declare no competing interests.
