## [Peer review file · Nature Communications]

REVIEWER COMMENTS

Reviewer #1 (Remarks to the Author):

Hong described the asymmetric β -pyridylations of enals through light-driven radical carbene catalysis. The reaction is inspired by their previous C–H pyridylation chemistry developed by the authors. This is based on the merging of radical carbene catalysis and the SET of pyridinium salt. The enantioselectivity is high level with various substrate combinations. The use of complex molecules derived from naturally abundant molecules is highlight for the present asymmetric system (Fig. 4). The enantiodiscrimination step is the radical addition between the chiral carbene-bound homoenolate and pyridinium salt. The enantiodiscrimination step that differentiates the two faces of the homoenolate radical is similar with Rovis's and Chi's report with nitroarenes (refs 47-49). However, the reaction with pyridinium salt is unknown. Overall, this project was nicely developed, and the text and SI are well-written and well-organized. The manuscript should be published in Nature Communications.^[1]

- Fig. 6: MeOH is missed.
- The following papers should be added to the corresponding parts:
<https://pubs.acs.org/doi/10.1021/acscatal.1c04153>
- Line 64: the value should be corrected. See the following paper, and add it.
<https://onlinelibrary.wiley.com/doi/full/10.1002/anie.202111988>
- How about the possibility of the formation EDA complex between I and 1?

Reviewer #2 (Remarks to the Author):

This manuscript describes an enantioselective beta-pyridylation of enals catalyzed by a chiral NHC catalyst. The reaction is believed to proceed through a single-electron transfer mechanism. Presumably, the transient homoenolate intermediate reduces NTs pyridinium salt to yield a beta-radical intermediate (Fig. 6, structure II). This radical adds to the electron-deficient pyridinium in an enantioselective manner, which upon subsequent electron-transfer and nucleophilic addition, yields the chiral beta-pyridyl ester. Reactions involving the beta-radical of homoenolate are quite rare, especially non-cyclization processes. The most notable examples are beta-hydroxylation reported by Rovia and Chi, respectively. The submitted manuscript accomplished an unprecedented enantioselective pyridylation, representing an important development. In addition, asymmetric radical addition to pyridine C4 is also highly challenging. Authors accomplished such a remarkable task, for the first time, using a chiral NHC as a

facial differentiator. The study was carried out with care and the presentation is quite clear. Publication is recommended.

Technical issues:

1. The mechanistic part is not adequate. The authors proposed a radical/addition mechanism. There is little evidence of a radical chain in the manuscript. Based on Ye and Huang's reports (ref. 50 and 51), the beta-radical tends to undergo radical-radical coupling with nucleophilic radicals. It is not clear how easy the beta-radical adds to an electrophile (pyridinium salt). The alternative radical-radical coupling pathway could also be operative, with the base promoting the final H-NMeTs elimination.

2. Can the author tether the two reaction partners using styrene?

3. It will be informative if the percentage of redox ester is included in the screening table.

Reviewer #3 (Remarks to the Author):

In the manuscript, Hong et al report an NHC-catalyzed enantioselective β -pyridylation of enals with N-NTs pyridinium salts. The reaction worked for a variety of enals and pyridinium salt, giving the desired products in good yields with good to high enantioselectivity. The C-selective functionalization of pyridine is a challenge. Thus the publication of the manuscript in Nat. Comm. is recommended after revisions.

(1)Scope. How about the reaction of β -alkyl enals?

(2)Mechanism. Radical clock experiment using β -cyclopropyl enal may be helpful to clarify the mechanism.

Reviewer #1 (Remarks to the Author):

Hong described the asymmetric β -pyridylations of enals through light-driven radical carbene catalysis. The reaction is inspired by their previous C–H pyridylation chemistry developed by the authors. This is based on the merging of radical carbene catalysis and the SET of pyridinium salt. The enantioselectivity is high level with various substrate combinations. The use of complex molecules derived from naturally abundant molecules is highlight for the present asymmetric system (Fig. 4). The enantiodiscrimination step is the radical addition between the chiral carbene-bound homoenolate and pyridinium salt. The enantiodiscrimination step that differentiates the two faces of the homoenolate radical is similar with Rovis's and Chi's report with nitroarenes (refs 47-49). However, the reaction with pyridinium salt is unknown. Overall, this project was nicely developed, and the text and SI are well-written and well-organized. The manuscript should be published in Nature Communications.^[1]

Response and Action: We thank the reviewer for the careful evaluation of our work and for the support.

-Fig. 6: MeOH is missed.

Response and Action: We thank the reviewer for catching this mistake. This error has been corrected as indicated by the reviewer in the revised manuscript.

-The following papers should be added to the corresponding parts:
(<https://pubs.acs.org/doi/10.1021/acscatal.1c04153>)

Response and Action: Following the suggestion, we have added this relevant reference (*ACS Catal.* 2021, **11**, 12886) in Ref 46.

-Line 64: the value should be corrected. See the following paper, and add it.
(<https://onlinelibrary.wiley.com/doi/full/10.1002/anie.202111988>)

Response and Action: We thank the reviewer for pointing this out. In accordance with the comment, we have corrected the values (-1.7 ~ -1.8 V) in the revised manuscript.

-How about the possibility of the formation EDA complex between I and 1?

Response and Action: We thank the reviewer for pointing this out. In accordance with the comment, we have now added the absorption data of **1a** and homoenolate intermediate in Figure S4 in the revised Supplementary Information to provide useful information to the readers. Although the reaction proceeded in dark conditions through thermodynamically favorable SET, irradiation with visible light facilitated more efficient conversion. Following the suggestion, we investigated the possibility of the formation of the EDA complex between homoenolate intermediate and pyridinium salt **1a**. As the

homoenolate form of the Breslow intermediate is hard to isolate and characterize, the reaction mixture was used in the absorption experiment. As shown in Figure S4, the homoenolate intermediate itself exhibited photon-absorbing nature in absorption spectra (400–550 nm range), and we observed only slightly increased absorption spectra in the mixture of pyridinium salt **1a** and homoenolate intermediate from **5c**.

In addition, please refer to Figure S3, which showed that the addition of sodium pivalate to pyridinium salt **1a** exhibited a significant enhancement in the absorption. These observations suggested that the pyridinium salt-pivalate complex effectively undergoes SET upon light absorption, and therefore the chain process could be reasonably postulated to promote single-electron oxidation of the homoenolate.

Reviewer #2 (Remarks to the Author):

This manuscript describes an enantioselective beta-pyridylation of enals catalyzed by a chiral NHC catalyst. The reaction is believed to proceed through a single-electron transfer mechanism. Presumably, the transient homoenolate intermediate reduces NTs pyridinium salt to yield a beta-radical intermediate (Fig. 6, structure II). This radical adds to the electron-deficient pyridinium in an enantioselective manner, which upon subsequent electron-transfer and nucleophilic addition, yields the chiral beta-pyridyl ester. Reactions involving the beta-radical of homoenolate are quite rare, especially non-cyclization processes. The most notable examples are beta-hydroxylation reported by Rovis and Chi, respectively. The submitted manuscript accomplished an unprecedented enantioselective pyridylation, representing an important development. In addition, asymmetric radical addition to pyridine C4 is also highly challenging. Authors accomplished such a remarkable task, for the first time, using a chiral NHC as a facial differentiator. The study was carried out with care and the presentation is quite clear. Publication is recommended.

Response and Action: We thank the reviewer for the careful evaluation of our work and for the support.

Technical issues:

1. The mechanistic part is not adequate. The authors proposed a radical/addition mechanism. There is little evidence of a radical chain in the manuscript. Based on Ye and Huang's reports (ref. 50 and 51), the beta-radical tends to undergo radical-radical coupling with nucleophilic radicals. It is not clear how easy the beta-radical adds to an electrophile (pyridinium salt). The alternative radical-radical coupling pathway could also be operative, with the base promoting the final H-NMeTs elimination.

Response and Action: We appreciate this insightful comment on the possibility of another pathway. Based on the previous reports, we believe that the reaction of beta-radical with electron-deficient pyridinium moiety is reasonable. Indeed, Rovis, Chi, and Ye (ref. 47, 49, and 50) reported the beta-radical readily reacts with electrophilic oxygen radicals and electron-deficient carbon radicals. In this context, the Huang group (In ref. 51) emphasized the novelty of their work as the first example of radical-radical coupling with nucleophilic alkyl radicals.

The reviewer also makes an interesting point, and we agree with the referee's view that the alternative radical-radical coupling pathway could be conceivable. On the other hand, after single-electron reduction of N-aminopyridinium salts through homolytic cleavage, pyridines and N-centered radicals are generally generated (Carreira and Togni, *Angew. Chem. Int. Ed.* **2020**, 59, 9264). In the case of electron-poor N-substituent (such as F and OCF₃) on the pyridinium salts, pyridyl radical cations are known to be formed as shown below, where the N-radical cations react with arenes to generate pyridinylated products. Based on these observations and literatures, radical/radical cross-coupling can be ruled out.

2. Can the author tether the two reaction partners using styrene?

Response and Action: This is a good suggestion. The reviewer requested to perform a three-component reaction by including styrene as a reaction partner. Following the suggestion, we have

investigated the reactions with styrene and other alkenes as well. In all these cases, however, we could not detect the formation of the three-component reaction adduct under the standard reaction conditions, and only β -pyridylated products were generated. With an activated alkene (enol ether), only a trace amount of an aminopyridylation product was detected as a side product (*Nat. Commun.* **2019**, *10*, 4117). These results indicate a polarity mismatch between the nucleophilic β -radical and olefins.

entry	Alkene	3a	3a'	Remarks
1		66%	-	
2		59%	-	Aminopyridylation product observed in trace amount
3		66%	-	
4		67%	-	

3. It will be informative if the percentage of redox ester is included in the screening table.

Response and Action: This is a good point. In accordance with the reviewer's suggestion, we have now added the percentage of redox ester in Table S2 in the revised Supplementary Information to provide useful information to the readers.

To place emphasis on this point, we have added one sentence in the revised manuscript as follows.

"when NHC catalyst **5f** was employed, the formation of the ester product was completely suppressed (see Table S2 in the Supplementary Information)."

Table S2 Optimization of the reaction conditions.

entry	NHC catalyst	yield [%]	er	redox ester [%]
1	5a	64	-	44
2	5b	47	78:22	37
3	5c	46	84:16	< 1
4 ^b	5d	55	85:15	25
5	5e	57	80:20	48
6	5f	58	89:11	< 1
7	5g	18	43:57	< 1
8	5h	< 3	41:59	< 1
9	5i	61	55:45	23
10	5j	53	75:25	30
11	5k	52	89:11	14
12	5l	51	79:21	10
13	5m	55	85:15	3
14	5n	50	89:11	21

Reviewer #3 (Remarks to the Author):

In the manuscript, Hong et al report an NHC-catalyzed enantioselective β -pyridylation of enals with N-NTs pyridinium salts. The reaction worked for a variety of enals and pyridinium salt, giving the desired products in good yields with good to high enantioselectivity. The C-selective functionalization of pyridine is a challenge. Thus the publication of the manuscript in Nat. Comm. is recommended after revisions.

Response and Action: We thank the reviewer for the encouraging evaluation and constructive comments on the manuscript.

(1) Scope. How about the reaction of β -alkyl enals?

Response and Action: We thank the reviewer for pointing this out. We attempted to use β -alkyl enals to probe the nature of the reaction. Unfortunately, we were not able to obtain the desired product and pyridinium salt **1a** was fully decomposed, probably due to the radical instability and decreased spin density. Further developments to apply this methodology to other substrates including β -alkyl enals are underway.

We have added a sentence in the revised manuscript to clarify the limitation on the scope of the enals. "When β -alkyl enal was subjected to the standard reaction conditions, the desired product was not formed."

(2) Mechanism. Radical clock experiment using β -cyclopropyl enal may be helpful to clarify the mechanism.

Response and Action: We appreciate the insightful comments raised by the reviewer. To comply with the reviewer's suggestion, we have investigated radical-clock experiments using β -cyclopropyl enal. However, we were not able to obtain any meaningful yields of β -pyridylation products under the standard reaction conditions. These new results have now been included in the revised Supplementary Information.

Previous radical-clock experiments using cyclopropyl-substituted enals (Chi, ref. 49) showed that isomerization or ring-opening of the cyclopropyl unit was not observed probably because the radical spin is more likely delocalized between the enal formal carbonyl carbon and the triazolium NHC unit.

REVIEWER COMMENTS

Reviewer #1 (Remarks to the Author):

In the revised manuscript, Hong and co-authors have responded reviewer's comments and questions properly, and given more data. Therefore, I recommend the publication of the revised manuscript in Nature Communications now.

Reviewer #2 (Remarks to the Author):

Most reviewers' comments were adequately addressed. Just two minor points. First, how much acceleration did light provide? The redox gap between the homoenolate and pyridinium salt is quite significant. Why $\text{Py}+\text{PivO}^-$ need to be excited? The authors should provide the dark experiment data for the optimized conditions. Second, I am just curious about the reactivity of 3-substituted pyridiniums? Such a substituent might hinder the addition of the homoenolate radical.

Reviewer #3 (Remarks to the Author):

The revisions are satisfied. The publication in Nat. Comm. is recommended.

Reviewer #2 (Remarks to the Author):

Most reviewers' comments were adequately addressed. Just two minor points. First, how much acceleration did light provide? The redox gap between the homoenolate and pyridinium salt is quite significant. Why Py+PivO⁻ need to be excited? The authors should provide the dark experiment data for the optimized conditions.

Response and Action: We thank the reviewer for pointing this out. Following the suggestion, we have added the dark experiment data under the optimized conditions in the revised manuscript (see entry 15, Table 1). When the reaction was conducted in the dark, lower yield was obtained (67% irradiated by blue LEDs vs. 53% in the dark). Since light-promoted reactivity and rate acceleration were observed, we propose an alternative mechanistic platform that involves a photon-absorbing pyridinium–pivalate EDA complex, which triggers the formation of an amidyl radical when irradiated with visible light, in turn, initiating a new radical chain pathway.

Second, I am just curious about the reactivity of 3-substituted pyridiniums? Such a substituent might hinder the addition of the homoenolate radical.

Response and Action: We thank the reviewer for pointing this out. The reviewer is correct that with C3-substituted pyridinium substrate, no desired product was detected probably because of steric effects. The results were included in Table 3 in the revised manuscript. We have added a sentence "The C3-substituted pyridinium salt was not a compatible with this method, and the desired product was not be obtained." in the revised manuscript.

REVIEWERS' COMMENTS

Reviewer #2 (Remarks to the Author):

The authors have adequately addressed the reviewer's comments. Publication is now recommended.